# Building Capacity for Community-Academia Research Partnerships by Establishing a Physical Infrastructure for Community Engagement: Morgan CARES

**DOI:** 10.3390/ijerph191912467

**Published:** 2022-09-30

**Authors:** Payam Sheikhattari, Emma Shaffer, Rifath Ara Alam Barsha, Gillian Beth Silver, Bethtrice Elliott, Christina Delgado, Paula Purviance, Valerie Odero-Marah, Yvonne Bronner

**Affiliations:** 1Prevention Sciences Research Center, School of Community Health and Policy, Morgan State University, 1700 East Cold Spring Lane, Baltimore, MD 21251, USA; 2Center for Urban Health Disparities Research and Innovation, Morgan CARES Community Engagement Core, Morgan State University, 1700 East Cold Spring Lane, Baltimore, MD 21251, USA; 3Center for Urban Health Disparities Research and Innovation, Morgan State University, 1700 East Cold Spring Lane, Baltimore, MD 21251, USA; 4Tola’s Room, 4212 Sheldon Avenue, Baltimore, MD 21206, USA; 5Hillen Road Improvement Association, Baltimore, MD 21218, USA; 6Department of Biology, Morgan State University, 1700 East Cold Spring Lane, Baltimore, MD 21251, USA

**Keywords:** community engagement, community-based participatory research, academia-community partnership, building capacity, underserved communities

## Abstract

Research partnerships between universities and communities following the principles of community-based participatory research (CBPR) have the potential to eliminate cycles of health disparities. The purpose of this article is to describe the process of establishing a community-campus network with a distinct mission and vision of developing trusting and successful research partnerships that are sustained and effective. In 2019, Morgan CARES was established to facilitate community engagement by founding a community center “within” a low-income residential neighborhood as a safe and accessible hub for creating a vibrant learning community. A community needs assessment and asset mapping was conducted and several necessary resources and services were provided to maximize networking opportunities, nurture innovative ideas and proposals, and provide seed funding. Lessons learned informed the optimization of a theoretical model that has guided the development and implementation of the program’s key components. By December 2021, Morgan CARES had recruited 222 community and 137 academic members representing diverse expertise from across Baltimore City. We also successfully established new partnerships and funded a total of 17 small community-academic awards. Although in its early stages, Morgan CARES has established a dynamic learning community following a conceptual framework that could guide future similar initiatives.

## 1. Introduction

Active participation of under-privileged communities in research has received increasing attention and funding over the past several decades for various reasons. Paulo Freire, the Brazilian educator, founded the basic principles of action research based on his endeavors aimed at empowering poor Brazilian workers through adult literacy education so they can be emancipated from the barriers that limited their potential [1]. By using symbols and concepts relevant to the workers’ lives, he was able to not only help them read and write, but also mobilize them to fight for their rights based on knowledge and insights gained throughout the process [1]. In the United States, the concept of Community-Based Participatory Research (CBPR) has been described as a methodological approach to bridge the gap between research and practice by developing community-owned solutions to address critical health disparities and social justice issues mainly through forming trusting relationships and fostering mutually beneficial collaborations between leaders and researchers inside and outside academia [2,3]. CBPR is guided by principles such as “recognizing community as a unit of identity”, “building on community strengths and resources”, and “promoting equitable partnerships” [1,2].

Following CBPR principles have proven to be challenging to achieve. Throughout the literature, a few distinctive features are described to distinguish CBPR studies from traditional academic research, such as: having a community advisory board or council overseeing the project [4] and, involvement of the community in designing, implementing, and disseminating the findings of the research [5,6]. Many CBPR partnerships, however, often fail to truly engage the community, achieve their intended transformative objective, and sustain the partnership beyond the funding period of the project [7,8]. For CBPR to be successful, previous studies have highlighted the importance of a few contextual factors and pre-conditions: building trustful relationships and shared understanding; balancing power and influence; negotiating rules of engagement; and shared governance [1,9,10]. 

Many educational materials and funding opportunities for developing CBPR studies currently exist, but few programs actively provide logistical (e.g., access to physical spaces and technology) or capacity-building opportunities (e.g., technical assistance, mentorship, training) to start and maintain effective and balanced partnerships [11,12]. While CBPR partnerships are often varied in their scope and strategic goals, when connected, they can form vibrant local networks to maximize networking, resource sharing, and mentoring opportunities.

Similarly, Morgan State University (MSU), an HBCU (Historically Black College & Universities) in Baltimore City, has worked intentionally over the past 20 years using the CBPR approach and has a history of working in the community with community engagement activities and programs. In 2010, MSU recognized those neighborhoods within a one-mile radius of the campus as the “Morgan Community Mile” and encouraged all academic programs and departments to actively collaborate with the residents and community leaders to enhance health, safety, economic status, and educational outcomes through innovative initiatives [13,14,15,16,17]. Despite challenges that have affected residents’ overall way of life, many Baltimoreans, similar to residents of other major cities, are taking charge. Reports show that Baltimore has become the central location of Black innovation and entrepreneurship, with 47% of small businesses being Black-owned, which plays a crucial part in community-strengthening for families (Baltimore Sun, 2020) [18].

Furthermore, through the CEASE (Communities Engaged and Advocating for a Smoke-Free Environment) project, a successful tobacco treatment program was developed involving a partnership between academic researchers and poor underserved neighboring communities of Baltimore City [19]. In addition, through ASCEND (A Student-Centered, Entrepreneurship Development model), a training program to increase diversity in biomedical research, small CBPR awards were funded, resulting in new research partnerships and a theoretical model [20]. CBPR built upon a strong foundation of community engagement, service, and volunteer work exemplified through the work of the CEASE and ASCEND programs shaped the foundation and need for establishing a learning community and network. 

The purpose of this article is to show how a trust-building needs assessment process, supported by an engaged steering committee, can be used to develop a sustainable community and academic partner-enhanced co-learning research environment using the Morgan CARES Model adapted from existing CBPR models. 

## 2. Materials and Methods

### 2.1. Setting

MSU’s Center for Urban Health Disparities Research and Innovation launched its Community Engagement Core, called the Morgan CARES (Community-Aligned REsearch Solutions) program, in 2019 to tackle the issues of community engagement using the CBPR approach through funding from NIH’s National Institute of Minority Health and Health Disparities (NIMHD) [21]. Morgan CARES is housed in a renovated historical building in an underserved neighborhood of East Baltimore. The facility serves as a hub for community and academic partnerships to produce high-quality, community-led projects that contribute to the understanding of health disparity issues and promote health equity. The main function of Morgan CARES is building connections, creating partnerships, and maximizing networking opportunities through recruiting members from diverse backgrounds. Morgan CARES provides technical assistance, seed funding to incentivize innovations, and support for partnerships as they embark on the implementation of projects and dissemination of the results. Morgan CARES *envisions* healthy communities where equity exists without struggle. Designed by researchers with extensive experience in public health, community-based, and clinical research, and community partners with extensive project implementation experience, Morgan CARES is serious about building capacity for community engagement while addressing health disparities. The overall function, leadership, and direction of Morgan CARES is governed by a Steering Committee consisting of 12 (4 resident leaders, 6 community-based, and 2 faculty) members with a broad spectrum of community experience and expertise. The committee provides programmatic oversight including strategic planning initiatives recommendations to ensure that Morgan CARES retains a balance between community and academic perspectives. Funding for the operation of the Center was partially sub-awarded to a non-profit organization (Fusion Partnerships, Inc., Baltimore, MD, USA) with strong community ties and relationships with equity-oriented local groups and leaders to serve as a fiscal sponsor [22].

Morgan CARES Community Center (MCCC) serves as the physically accessible space for creating a dynamic learning community where community and academic members network, share knowledge and resources, and work toward eliminating health disparities while producing new generalizable knowledge. MCCC also provides a wide range of free services including office space for meetings, a kitchenette, a space for CBPR projects that involve youth, and access to computers and printers for community use. Partnerships are formed and strengthened through research training, technical support, workshops, classes, and community events and activities. 

The community center is surrounded by four neighboring underserved urban communities, depicted using a Geographic Information Systems (GIS) mapping tool (map in Table 1), all of which have faced historical health and investment challenges. To assess the neighborhoods’ social determinants of health and health outcomes, we used the Baltimore City Neighborhood Health Profile 2017 (Table 1). The total population of over 32,000, is about 90% Black. Social determinants of health metrics, such as educational achievement, median household income and percent poverty are not optimal when compared with those of Baltimore City overall. 

Many households live below the poverty level and have high rates of unemployment. Only an average of 9% of the population 25 years of age and older have completed a college degree (or more). In these same areas, health-related problems are entrenched with the highest rates of mortality due to heart disease, cancer, and HIV. Life expectancy at birth is between 66–70 years compared to 73.6 years for Baltimore City. These documented snapshots of the state of health in East Baltimore shown here are not relevant to just this region of the city; similarities are found throughout low-income neighborhoods in Baltimore City. With this knowledge, Morgan CARES believes that working toward the goals of health equity, by partnering with community stakeholders across Baltimore City, is not only possible but necessary.

### 2.2. Morgan CARES Model

The Morgan CARES Model is designed to provide a framework for conducting CBPR work in communities to address health disparities and is guided by Wallerstein’s CBPR conceptual framework on CBPR partnership success [3,23]. Figure 1 depicts the Morgan CARES structure for developing a well-aligned community engagement and partnership approach. This model uses a sequential cyclical exploratory methods design and follows a multi-stage process to develop a measurement tool to track achievements and assess long-standing CBPR partnership success for any given project. Morgan CARES enhances partnership success by providing resources and activities at each stage of development that will aid in implementing projects, ultimately leading to a productive learning community.

The Morgan CARES Model has five stages: (1) **Connection** is the initial stage where academic and community members learn about each other and their research interest. For example, a first-time member would complete a form that captures the reasons for their interest in joining the network. From there, all members have access to information sessions to learn the benefits of joining Morgan CARES, and introductory training to gain knowledge of how Morgan CARES aims to work towards addressing public health challenges in local communities, and community events. (2) **Partnership Development** occurs when participants commit to increasing their engagement from the “Membership” level to the “Partnership” level. The Member will provide information about their research interest and expertise. The Morgan CARES administrative team uses this information to match the partner, (academic or community) with individuals who have complementary attributes and interests. It is within this stage that partners receive supportive services including targeted communications for advanced training, networking opportunities, and discussion forums. Though individuals join with different levels of skills, we provide space for reciprocal exchange and bidirectional learning. (3) **Innovation.** Once connections with potential partners have been made, matched partners (community-academia) begin to discuss project ideas, formalize their commitment to work together, and design a research project. Partners would also attend advanced targeted training and other specific skill-building opportunities that support the development of their proposal including consultation on grant-writing, proposal preparation, and proposal revision. Partners would receive ongoing support services for their project needs including technical assistance and seed funding opportunities. (4) **Collaborative Action.** It is expected that funding (or other resources) has been secured and partnerships have initiated project activities. Partners have access to MCCC at low or no-cost to support project implementation and activities including technical assistance, and logistical support to further sustain their collaborative activities. (5) **Outcome and Impact.** Analyses conducted to determine outcomes and research results are formalized into reports that can be shared. These results may be made available to various stakeholders, such as the scientific communities and non-scientific communities, policymakers, students, faculty members, and other interested parties. In addition, partners may receive supportive services such as technical assistance and targeted learning activities to help sustain projects and partnerships including professional writing, grant sourcing, data management, and linkage to additional resources or services. During this stage, partnership teams can use additional support to report their findings. Following dissemination, seed-funded projects have the potential to mature into larger projects that promote changes in health policy and programming, as well as impact health behaviors within communities.

### 2.3. Community Needs Assessment

As the first step in connecting and developing strategies for health and wellness projects, Morgan CARES began with a community needs assessment to set the stage for building a trusting, mutually beneficial relationship. The first needs assessment conducted was specific to developing the activities to support the Community Award Initiative. There were four (4) key informant interviews, and three (3) focus groups conducted with community residents and other stakeholders (*n* = 25) within a mile radius of the MCCC. The information gleaned from these efforts informed the initial development of activities to support Morgan CARES Community Award initiative. The second needs assessment was conducted to identify the programming needs for groups doing work in the community. Several community-based organizations, faith-based groups, and other resident leaders participated in the four (4) key informant interviews and two (2) focus groups (*n* = 29). Interviews and focus groups were conducted based on the type of work individuals were involved in and facilitated by MCCC staff and researchers. Examples of interview questions included “Can you tell me about yourself and a brief history of your work in the community?”; “What barriers or challenges does your organization currently face?” and “What does successful community engagement look like?” All interviews and focus groups were conducted virtually utilizing the Zoom^®^ platform. Interviews were documented by hand-written notes, audio recordings, and transcriptions using Panopto.

### 2.4. Morgan CARES Community Award Program and Projects

The Community Award is an initiative that provides seed funding of up to $2000 (each) for up to ten (10) innovative, small, health-related projects each award cycle. The goal of this award is to support the formation, strengthening, and maintenance of community-campus collaborative partnership teams in the production of meaningful initiatives that could be scaled to improve health equity in Baltimore City. The Community Award process involves community leaders and academic experts connecting and partnering that follows a sequential process based on the Morgan CARES model that first begins with joining the network to become a “Member” (Figure 2).

#### 2.4.1. Connection and Partnership Development

According to the Morgan CARES model, the initial step is about providing representatives from nonprofits, academia, and faith- and community-based organizations with opportunities to connect with others in the network and target issues that directly impact their community. By joining the network as a “Member”, individuals will receive updates about Morgan CARES from regular e-newsletters, email announcements, information and resource sharing, opportunities for funding, access to the Morgan CARES Community Center space, events and activities, mentorship, and support with external projects (trainings, workshops, etc.). Participants could express interest in increasing their engagement from the “Membership” to “Partnership” level. The partnership includes community partners getting linked (matchmaking) to potential academic partners based on common research interests for the purpose of collaborating on health-related projects, programs, and initiatives. A partner receives benefits such as priority access to training sessions, workshops, the MCCC, professional development, and eligibility for the Morgan CARES Community Award. Partnership teams include co-leads: at least one Community Lead and one Academic Lead (from MSU).

#### 2.4.2. Innovation and Collaborative Action

Once community-academic partners have identified a research problem, the partners work together to co-develop an innovative project proposal that can research solutions to improve health. In this stage, the team develops a project plan that explains how funding would be used to address these needs. The specific roles of each partner are also discussed in addition to the sequential steps needed to achieve the aims and the expected impact on community health. All project teams have access to training, seminars, workshops, technical assistance, and consultations provided by Morgan CARES to support the development of proposals. All proposals must include components based on co-identified needs such as the project title, timeline, aim, type of project, focal community, description of health problem, activities and timelines, expected impact on community health, amount requested, budget, additional funding, and ethical considerations. Eligible applications must be submitted by a community-academic partnership, have a project timeline of 1 year or less, contribute to the overall improvement of health, target a specific group or community, be committed to disseminating the results, and expand the project and partnership. Partners submit proposal applications for review by the submission deadline. All applications go through a three-step review process. The first step is the Administrative Review by the Morgan CARES Administrative team, an initial review to check each application for completion. Only completed applications move forward to the second review step, the External Review. At this step, one or more expert reviewer(s) in a relevant field (who are not affiliated with the Morgan CARES Program) examine and score the proposals. The top-scoring proposals are provided to the Steering Committee Review Task Force, who then review and rank the proposals they feel should be funded. Following a discussion among the Task Force members, the top proposals are selected for funding by the Morgan CARES Community Awards Program.

#### 2.4.3. Outcome and Impact

At this step, project funding has been secured, and community-academia partners have commenced implementing the project activities. Teams engage in ongoing training, workshops, seminars, and networking activities that support the progress of the project’s success. Each team submits a collaborative Mid-Project Report that gives updates on the project status and the summary of progress. A Mid-Project Partnership Evaluation survey is also completed individually, which assesses aspects of the partnership experience thus far. Lastly, upon completion of project activities, project teams evaluate the outcomes and impact of their project. This stage also includes steps of identifying funding sources that support the continuation of the project and plans for sharing project outcomes with the community through dissemination efforts, activities, and materials. All awardee teams complete a Final Project Report and Partnership Evaluation that addresses skill development throughout the process describes any needs for improvement and plans for sustainability and future collaborations.

### 2.5. Metrics Measure Evaluation Designs

To determine effectiveness, metrics were developed based on the Engaged for Equity (E^2^) CBPR Conceptual Model to evaluate the outcome and impact of the community award initiatives overall [23]. This model captures the process and outcomes of CBPR and community-engaged research projects with the goal of promoting practices with equal partnering that will strengthen and in turn produce long-term outcomes that transform communities and improve health equity [23]. Data were collected from the mid-project and final project progress reports of the community award recipients [24]. Using the E^2^ tool, data included indicators for success for each stage measuring: (1) capacity (ability to make connections, partnership capacity, and community history); (2) commitment to collective empowerment and relationships (partnership principles, community fit, influence in the partnership, and collective reflection); (3) relationships (leadership, dialogue and listening, conflict resolution, and trust); (4) community engagement in research actions and synergy (background and design, analysis and dissemination, and community action); and (5) outcomes to measure partner and partnership transformation (personal benefits agency benefits, community power in research, sustainability) and projected out projected outcomes (policy, community integration in research, social transformation, and health improvement [24]. Evaluations were organized into two categories: “The Project” and “The Partnership”. There are three metrics for the project report, including progress toward aim, community involvement, and outcome and impact with corresponding indicators for each metric. Questions under each metric were derived from self-reports submitted by the partnership and were completed collaboratively between all partners. Example questions included: “What progress has been made toward achieving the project objectives?” “What have you learned thus far?” “What challenges, opportunities, or barriers have you encountered thus far?” and “What are your plans for completing your project within the remaining allotted time?” Specifically, “the partnership” explored various aspects and influences on successful partnership formation and maintenance and was derived from the Partnership Evaluation form each partner completed individually. There are five metrics including equity, conflict resolution, involvement, commitment, and benefits, and corresponding indicators for each metric. Example evaluations included “Did you have a relationship with your partner prior to this project”, “I have learned from my partner(s)”, “Community (or Academic) partner have been involved in all stages of the process thus far”, and “My partner(s) and I generally agree on the mission and goals of this project”. All partnership evaluations involved qualitative and quantitative data collection.

## 3. Results

### 3.1. Identifying Community Challenges Perceived by Community Stakeholders

In integrating community perspectives into all stages of research and connecting with neighbors centered around the Morgan CARES Community Center, it was necessary to gain an understanding of the work being done, the priorities, successes, challenges of the community, and potential opportunities for collaboration. This process aided in understanding the historical context of the surrounding community and identified resources needed to develop a sustainable initiative that would improve health equity. Among several topics discussed in the focus groups and interviews during the second needs assessment, the majority of the community stakeholders identified barriers and current challenges their organizations faced that they believe were critical to addressing health equity concerns in their community (Table 2). Challenges that affected their organizations included organizational capacity in terms of needing community efficacy, having access to resources, integrating equity, sustaining data, sustaining funding, and lack of leadership training. Oftentimes, these challenges were temporary setbacks such as securing funding, whereas in other cases, having sustainability and mobilizing were long-term issues. Understanding stakeholders’ perspectives, concerns, and interests, Morgan CARES discussed solutions to address these barriers to build capacity and maintain sustainability through sources of support such as providing funding opportunities, leadership training, and collaboration through community engagement.

### 3.2. Morgan CARES Network Services and Community Award Initiatives

Morgan CARES actively serves as the foundation for establishing sustainable infrastructure that forms strong long-lasting partnerships among community members working to improve health throughout Baltimore. By December 2021, 222 community and 137 academic stakeholders across Baltimore City and several counties in Maryland had joined the Morgan CARES network. Review of their descriptions illustrates how they are directly impacting Baltimore communities (Figure 3). Community members who attended “orientation” or connection activities received introductory training, list a of community events, and health concerns information. Of the members, 61 (~16%) became Partners with Morgan CARES’ mission of advocating for health, wellness, and health equity through Community-Based Participatory Research. The rewards of collaborations between community leaders and academic leaders in this critical health concern are the possibility of increasing and strengthening the breadth and depth of understanding health challenges and identifying solutions that best address them [25]. 

Important to note, several community and academic (MSU) leaders had pre-existing relationships, while the majority of community leaders required linkage and matchmaking services to identify connections with academic faculty and students at MSU. Additionally, over 75% of community leaders already had an idea for their project. Workshop and consultation sessions were provided to support partnerships in every stage: proposal development, seed funding award, implementation, and findings dissemination. Over the past 2 years (2020 and 2021 fiscal years), Morgan CARES (including 2 research assistants, a peer advocate, a program coordinator, 2 faculty members and external partners) conducted over 187 activities with 969 persons in attendance (Table 3). Group check-ins were scheduled regularly, and one-on-one support was provided. Despite the challenges presented by the COVID-19 pandemic, services and activities were available for all stages and hosted virtually. 

The seed funding from Morgan CARES created a real impact in our local community by allowing ideas to expand, grow, and essentially come to life. Overall, 46 complete proposals ((Cohort 1 (2020–2021) and Cohort 2 (2021–2022)) were submitted and went through the competitive review process detailed in the “Methods” section. Recommendations were made based on the outcome of the review process, and projects were either recommended for funding pending “minor revisions” or “major revisions”, or they were “recommended for resubmission, pending major revisions”. Seventeen (17) projects received $2000 of funding to carry out their projects, though two (2) were discontinued. Partnerships whose projects were not funded and “recommended for resubmission” received one-on-one assistance to help develop proposals further based on reviewer comments. Three (3) projects that were recommended for resubmission from Cohort 1 and were resubmitted in Cohort 2 received funding. As described in Table 4, the funded projects can be categorized as (1) “health promotion and education”, which involves the dissemination of knowledge and information. For example, one project developed COVID-19 educational materials for teachers to use to prepare students for the return to in-person classrooms; (2) “Intervention”, which involved project teams introducing new approaches. As an example, one project explored how ultra-violet lighting can be used as a germicide to clean high-touch surfaces; or (3) “Evaluation”, which involved assessing existing programs and identifying opportunities for improvement. Another project assessed the feasibility of a community walk-through theatre, to determine the possibility of improving emergency preparedness and health communications among residents in the community. Funded community-academic projects involved faculty, staff, and graduate students from MSU who provided mentorship throughout the duration of the projects. Completed projects were presented at the Morgan CARES Inaugural Symposium in December 2021.

## 4. Discussion

There is an abundance of community-engaged research examining health disparities in underserved communities and primarily focusing on reducing disparities through conducting translational research. There are very few programs, however, with explicit missions of enhancing the capacity for conducting CBPR and facilitating research partnerships with underserved communities [11,12]. Partnering with organizations beyond academia through strategic collaboration for research and mobilizing underserved community engagement is critical [26]. In this article, we began with the essential CBPR principle of trust building by conducting a needs assessment to learn from the target community what their wants are. Our community engagement core, Morgan CARES, was established to build cohesive support between ongoing community projects to synergize their current efforts in the community. Components of effective community engagement require interaction between a diverse group of community members which typically creates more opportunities for learning and sharing knowledge to build a healthy community [26]. By adapting elements of existing CBPR models, our work allowed the Morgan CARES Model to evolve. The Morgan CARES five-step model provides co-learning, training, and technical support opportunities that can remove barriers for research project development and implementation. The qualitative research conducted for this study adapted existing instruments whose reliability and validity have been established based on the work of Wallenstein et al. [23,27].

Morgan CARES has had two unique outcomes that can be useful in moving CBPR methods forward: (1) we created a well-constructed and appointed community learning space where both community members and academics can meet in a common place; (2) and we developed a Community Award program that provide support activities tailored to building community-academic partnerships. CBPR is based on a trust-building process [28], and Morgan CARES community accessible space is that infrastructure. To date, the Morgan CARES Community Center has played a vital role in creating a true sense of community where groups work together in a common place for a common purpose [29]. In most cases, universities participating in CBPR do not have this entity. The Morgan CARES community center is uniquely placed in a community that has legacy neglect resulting disparities in all determinants of health. However, people in these communities have risen beyond the challenges faced and have developed something meaningful to build and create a way to give back to their communities in hopes of reviving underserved communities [30]. As new neighbors moving into a new neighborhood typically do a meet and greet, we connected with our neighbors surrounding the community center which was a critical step for informing the community of the work Morgan CARES does and its mission of eliminating health disparities and most importantly understanding the real needs of the community. As stated by the Centers for Disease Control and Prevention in achieving health equity, a targeted understanding of a community’s environment and perspectives is crucial in understanding the needs, promoting strengths, and directing resources toward the community so that everyone obtains “full health potential” [31,32]. The information obtained from the community needs assessment from community stakeholders provided us with points for discussion related to their organization’s strengths, needs, barriers, and resources. The most significant need addressing health disparities of the greatest concern to the community was identified and prioritized. While the community stakeholders had over ten years of experience working in the community, several saw a dire need to build their organizational capacity to make a successful change in the community. Many concerns surrounded sustainability, leadership training, funding opportunities, etc. (i.e., neighboring organizations perceived sustaining partnerships to be challenging due to a technological gap and lack of adequate infrastructure). While these findings suggest that other organizations in this region may also be facing similar issues, Morgan CARES meets this need by providing services and resources to address such barriers. Additionally, information derived from this process confirmed and supported the direction to include in the process of building community capacity and maximizing community strengths and assets to meet unmet needs and eliminate health disparities [26,33]. 

To effectively make a difference through CBPR, key community stakeholders must be properly engaged. Stakeholders including community members, nonprofit organizations, activists, medical professionals, and policymakers that focus on health equity are required to be equipped with a better understanding of why there are continued cycles of health disparities in underserved communities [11,12,34]. It absolutely takes a village, a community of people, with like minds to achieve the goal of health equity. Community stakeholders across Baltimore who joined our network as community partners brought such diverse experiences and skills to the Morgan CARES learning community. The widespread distribution of partnerships in Baltimore creates an idea of the impact Morgan CARES will have on the city as a whole. Engaging diverse stakeholders can help identify, prioritize, and integrate community and academic needs and resources, and help partners align around a shared vision [29,35]. Morgan CARES used these as examples to enrich the learning process, and how all these different backgrounds can come together to reach a common goal. For example, the majority of Morgan CARES partners are affiliated with community-based organizations, while others are either affiliated with academics, health care, government, or a local agency. 

Morgan CARES has used its model to create the infrastructure needed to implement small grants with academic/community partners. (Tools created for this process are available upon request). A key implementation strategy leading to the success of our program included following partners through all steps of research. We watched partnerships evolve into what the Morgan CARES team believed was possible, building collaborative relationships between community and academia. We carefully monitored partnerships’ progress and provided the necessary resources, relevant to each stage of the Morgan CARES model, as such to ensure accountability and the impact of relationships. Partnership evaluation was also an essential part of the Morgan CARES program as it helped in strategic planning, redefining strategies, taking appropriate actions based on recommendations, and improving resources to build capacity and sustain successful CBPR [36]. We chose to do mid and final partnership evaluations and reports using quantitative and a formative report method, respectively. Mid-project evaluations are a means of reinforcing partnerships and the process of collaboration and ensuring trust between partners and that the project is going well. Using both methods of evaluation supported strengthening services and/or workshops during the initial cohorts and for cohorts that follow. Integrating learning services and resources as a part of core activities within the Morgan CARES network was well received by community-academia partners. Many agreed that this partnership helps individuals grow and learn from one another and create space for sustaining long-term relationships. While Morgan CARES successfully initiated its Community Award Program to support community-academic partnerships, two (2) projects from the first cohort were discontinued due to not having access and or feasibility to follow through with proposed plans such as challenges with a commitment between the partnership. These are important lessons learned that will be used to make our project stronger in the future. Additionally, COVID-19 posed significant challenges related to gathering in-persons, a core component of having the Community Center as a place to physically gather. Our evaluation led to the following changes: As a result of the low participation in supportive services for the first cohort, we revised the types of services needed to a 2-day intensive workshop centered around partnership and CBPR that included input from diverse individuals and experts which resulted in a better outcome for the second cohort. An example of an effective community/academic partnership is the MSU NIH-funded CEAL (Community Engagement Research Alliance) project, emanating from Morgan CARES with the goal of providing communities with accurate information so that residents can make their own informed decisions to protect themselves and/or their family members during the COVID-19 epidemic. This successful grant application was only possible because we had existing strong community partnerships. An important lesson learned from working with five partners from the Morgan CARES network is that the COVID funding made possible the opportunity for us to provide substantial, much-needed resources to enhance the work that the partners were already doing to help families cope with COVID-19 [37].

Other challenges for CBPR work include difficulty in finding faculty to commit to CBPR due to a lack of funding. For the successful implementation of this program, small grants should be increased considerably to allow our community partners to be adequately compensated for the time that is put into these projects. In this way, the faculty would be willing to make this investment. In addition, seed funding was used to create a pathway that aid in building the community-academic relationship; however, there should be a pathway for continued funding that is guaranteed to support the formed partnership. Despite these challenges, our work is moving in the same direction as NIMHD by opening doors for this work to move and can be funded appropriately.

## 5. Conclusions

In conclusion, this article describes the relevance and significance of the Morgan CARES community center and how it plays a crucial part in creating the opportunity to build research projects that can promote health equity in underserved neighborhoods in our communities. Although still in its early stages, the Morgan CARES approach is a successful model and provides the groundwork for other community engagement cores to follow in the development of critical strategies for integrating sustainable support for CBPR. Morgan CARES has a well-appointed space for people to meet and has created the means of building trusting relationships within our neighborhoods, despite historical challenges, and fostering a co-learning environment. The disparity in health based on where you live is worldwide and has been attributed to the inequitable distribution of resources necessary to support health and wellbeing. Therefore, developing an infrastructure within communities that focus on fostering relationships is an important first step in moving toward solutions with key community members to enhance overall health outcomes.

## Figures and Tables

**Figure 1 ijerph-19-12467-f001:**
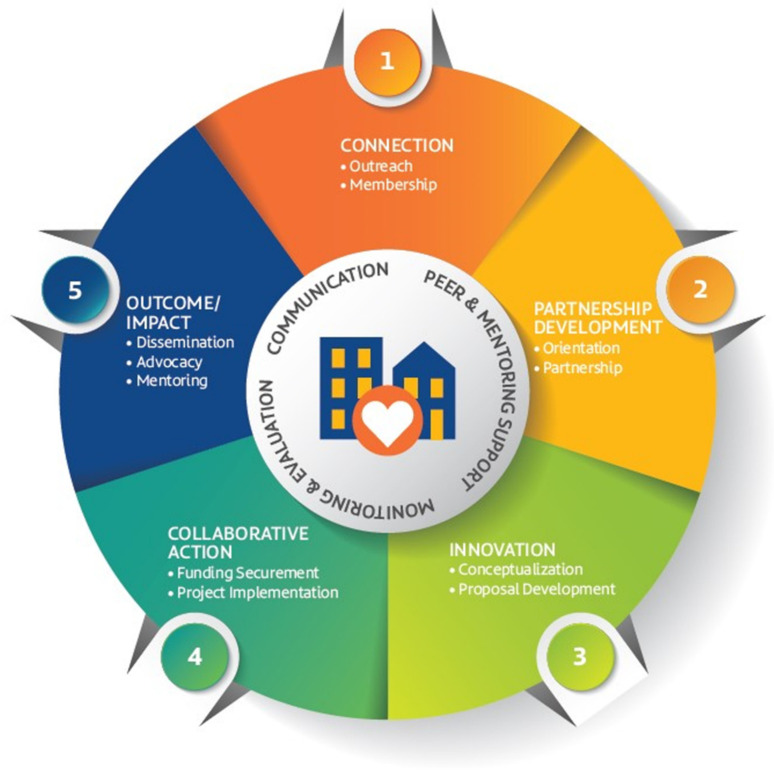
Morgan CARES Model.

**Figure 2 ijerph-19-12467-f002:**
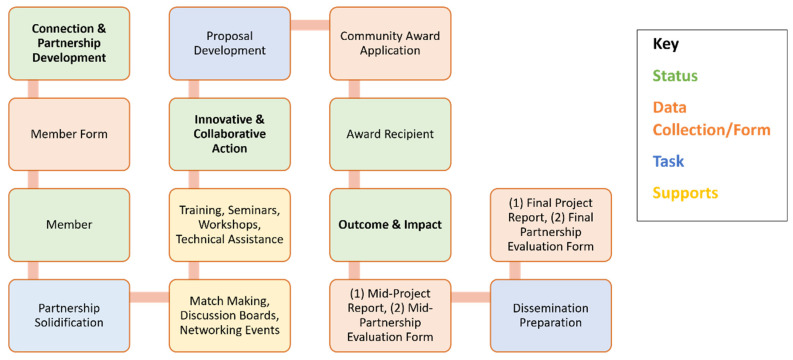
Morgan CARES Community Award Process.

**Figure 3 ijerph-19-12467-f003:**
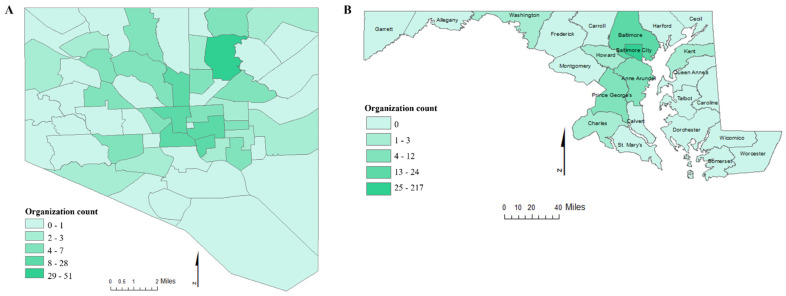
GIS map depicting the distribution of Morgan CARES Members’ Organizations in (**A**) Baltimore City and (**B**) across counties in Maryland.

**Table 1 ijerph-19-12467-t001:** Neighborhood Composition and Social Determinants of Health in East Baltimore *.

	Clifton-Berea	Greenmount East	Madison/ East End	Oldtown/ Middle East	Baltimore City
**Resilience in Baltimore (Entrepreneurship)**					
47 % of Small Black-Owned Businesses					
**Total Population**	8413	7691	7204	9285	622,454
**Education**					
% Adults with High School Diploma or less	63.3	66.7	72.0	68.1	47.2
% Adults with College Degree or more	7.7	8.2	6.3	15.0	28.7
**Socioeconomic Environment**					
Median Household Income	$25,738	$23,277	$27,454	$14,105	41,819
% Income < $25,000	44.8	52.4	48.3	66.5	32.2
% Unemployment Rate	17.4	24.7	26.4	20.7	13.1
% Family Living in Poverty	30.2	33.8	45.2	60.0	28.8
**Health Care Insurance**					
% Adults without Insurance	10.0	12.2	15.5	13.2	11.7
% Children without Insurance	3.4	0.6	6.0	2.8	4.4
**Life Expectancy at Birth (years)**	66.9	67.9	68.9	70.4	73.6
**Top Leading Causes of Mortality ^#^**					
Heart Disease	27.7	42.3	41.2	35.3	24.4
Cancer (all types)	24.9	37.6	44.7	30.5	21.2
Stroke	6.9	7.0	12.7	5.1	5.0
Diabetes	2.8	5.0	4.7	4.1	3.0
HIV	2.7	3.8	3.7	1.3	1.8
Drug- and/or Alcohol-Induced	8.3	8.1	6.9	8.4	4.4
Homicide	10.4	2.6	6.4	4.0	3.3
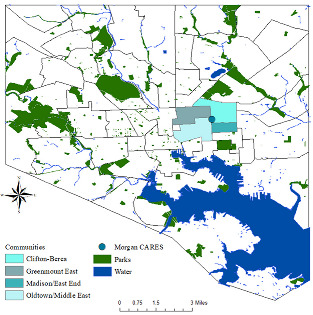					

* Source: Baltimore City 2017 Neighborhood Health Profile. ^#^ Mortality Rate per 10,000 residents.

**Table 2 ijerph-19-12467-t002:** Perceived Challenges by Community Stakeholders and ways Morgan CARES provides Solutions.

Barriers and Challenges Organizations Currently Face	Issues	Solutions	Example Quotations
**Organizational Capacity** defined as **mission, vision,** **services, initiatives, capacity, sustainability,** and **funding**	“historically not had access”	“Morgan CARES meeting space provides an opportunity for individuals to connect with others”	“I was wondering what’s the mechanism that you are thinking about for making that real and for making sure that people are even aware that there are some people who are going to be in the know and have access to, but, but then there are others might not know that this opportunity exits?”
	“Sustainability”	“funding opportunities”	“Hard to sustain because it is very expensive.”
	“technological gap” “how to sustain data collection”	“leadership training”	“We have people within the community that are resident researchers that are currently right now going into the community to figure out the needs and getting data and really making sure that the data we are gathering is what the community wants. But we are trying to figure out how to sustain that data.”

**Table 3 ijerph-19-12467-t003:** Morgan CARES Support Activities and Services (2020–2021).

Stage	Activities	Sessions	Attendees
**Connection & Partnership Development**	Outreach, Networking, Information Sessions, Linkage, Matchmaking, Introduction Training	83	406
**Innovation**	Orientation, Technical Assistance, Partner Consultation	65	244
**Collaborative Actions**	Skills Training & workshops, Project management support	23	205
**Outcome & Impact**	Project Evaluation, Consultation with experts, Dissemination Support	16	114

**Table 4 ijerph-19-12467-t004:** Summary of Community Campus Award Projects.

Project Title	Principal Investigators (Community—Academic Partnerships)	Initiative
STOP-OD: Prevent Deaths from Opioid Overdose	Haygood, J. (Applications Operation, LLC)—Estreet, A. (MSU, Social Work)	Mobile app for community health workers and family members to access resources and alert first responders in cases of potential OD’s
Safer Schools	Gordon, S. (Cool Green Schools)— Gibson, S. (MSU, Education and Urban Studies)	COVID hygiene and cleanliness educational materials for students in preparation for returning to in-person learning
Purple Light Project	Najee Ullah, M. (FullBlast STEAM)— Lewis J., Balraj, D., Ekpew, A. (MSU, P.I.s)	Testing UV light as a germicide on high-touch surfaces
Mid-Day Check In	Smith, D. (SolFlowers)— Holland, J. (MSU, Family and Consumer Sciences)	Talking sessions about mental health, wellness, nutrition over a healthy meal
Ivanhoe Valley Garden	Govan, N. (Wilson Park Northern Neighborhood Assoc)—Holms, K. (MSU, P.I.)	Beautifying neighborhood lot for neighbors to grow healthy foods, trying to establish a long-term program for volunteers to tend to the plots
Food Life Series	White, A. (I AM Whole, Inc., Baltimore, MD, USA.)— Brown, E. and Peterson, J. (MSU, Nutritional Science)	Developing educational sessions to help MSU students become food secure, and avoid becoming food insecure in the future
ECBB Emergency Preparedness Program	North, J. (Empowering Communities Block by Block)—Rowel, R. (MSU, P.I.)	Assessing the feasibility of using the community walk through theatre as a means of delivering emergency preparedness and health communications to neighborhood residents
Older Women Embracing Life (OWEL)	Richards, G. (OWEL)— Weaks, F. (MSU, Health Science)	A storytelling film about the oldest cohort of women living with HIV/AIDS in Baltimore City
The MD Healthy Workplace Task Force	Glover-Kerkvliet, J. (Baltimore Job Hunters Support Group)—Page, R. (MSU, P.I.)	To develop an 8-week training on the diagnosis, treatment, and prevention of workplace bullying and mobbing
Dyslexia Awareness Campaign	Winston, W. (So All Can Read)— Gibson, S. (MSU, Teacher Education and Professional Development)	Spread dyslexia awareness, educate parents, remove stigma, and reduce financial burden.
Educational x Tech Training Program (EDxTech)	Best, A. (Baltimore Tech Hub)— Wright, M. (MSU, P.I.)	Provide educational resources within communities while using tech to elevate understanding
Cherry Hill Food System Assessment	Jackson, E. (Black Yield Institute)— Walker, K. (MSU, P.I.)	To address unequal distribution of land and food access by surveying Cherry Hill residents’ perceptions of the food system as food apartheid
#JustDont: Youth Anti-Litter and Art Advocacy Movement	Delgado, C. and Sharif, N. (Tola’s Room)— DePaolo, B. (MSU, Fine Arts)	Pilot project to activate conversations and action around creating "litter free zones" in the Belair-Edison community
Light Within the Margins	Garcia, M. (Light Within (& CWTT))— Reeder, (MSU, Visual Arts)	To address the impact of ACES primarily through trauma informed creative workshops and storytelling
CAMMRAD Police App	Doswell, J. (Juxtopic)— Sinclair, M. (MSU, Social Work)	To eradicate the endemic of police violence against African American males in Baltimore, MD

## Data Availability

Not applicable.

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
