# Peer review of "Building Capacity for Community-Academia Research Partnerships by Establishing a Physical Infrastructure for Community Engagement: Morgan CARES"

_ijerph, 2022, doi:10.3390/ijerph191912467_

Round 1
Reviewer 1 Report
Interesting manuscript and I enjoyed reviewing it. Just some minor things that should be address by the authors:
· The goal of this manuscript was not clearly stated. It should be clear what contribution this study brings to the specific subject field. Add a sentence in the abstract and Introduction section to clarify the purpose of the manuscript.
· The reliability and validity of the research / model has not been adequately stated.
Strengths:
1) The title is descriptive and concise.
2) The manuscript provides an original contribution.
3) Logical conclusion has been drawn.
4) The referencing is correct, appropriate, and current.
Weaknesses:
1) The introduction and/or literature review do not adequately describe the goal/purpose of the study. Add a sentence or two to clarify. Suggestion: Add a sentence in the abstract and introduction.
2) Reliability and validity of instruments / data have not been stated. Add a sentence to report on the reliability and validity of the instrument (model).
Reviewer 2 Report
This is a very well written article describing a academic-community initative with a thoughtful community engagement process and funding mechanism to foster partnerships. I provide detailed comments below - the majority of which are suggestion I think will help to provide guidance to readers on how they could build a similar initative at their university.
One larger consideration I would suggest is to engage community co-authors on th revision of the paper. This is one of the core activities of CBPR and it would add great value to have the community perspective interwoven. Community co-authors might be steering committee members or someone from Fusion Partnerships that were funded on the project.
Abstract:
Line 28 - I suggest listing the number of academic and community members separately
Introduction:
Strong background on CBPR at the beginning of this section!
Lines 63-66: Could you add a citations for this claim? All CBPR projects that I l know of have had some sort of capacity building opportunities involved. Are there other logistical examples? Do you have a sense of how often tech or space would be needed for CBPR projects?
Lines 73-77: Could you add any citations - papers, websites, news articles - that can be added to share more detail about the 2010 "Morgan Community Mile" initative?
Materials and Methods:
Lines 106-109: Could you add details on the size of your steering committee and an appendix with the names and background of the steering committee members? This type of detail could be helpful to readers who would like to build something similar?
Table 1 is excellent - I like the side by side map and demographics. The only thing that was unclear to me was why "47% of small back-owned businesses" was included. It is clear in the text, but if you don't have it by neighborhood it isn't clear what it means in the table. You might also consider including a comparison to Baltimore overall or US average in the table to demonstrate the contrast.
Section 2.2 - Does the model work the same for community and academic members? Would they all complete the same interest form? Attached the same information sessions? Please describe how these activities might be different at each stage.
Section 2.3 - It isn't clear how the two phases of the needs assessment are different from one another. It looks like they all happened over the course of the same year. Were different questions asked? Did the first round inform the second? How were the results used? Who conducted the interviews and focus groups? Why did you include both interviews and focus groups? How were they different?
Section 2.4 - It would be great to learn more about the Community Award program. What is the duration of the funding? These are small awards - what needs to be included in the proposal? Are there community members involved in the external review of the proposals? What are the review criteria?
Section 2.5 - Could you include the specific items asked in the evaluation? Are can taken from existing valided measures?
3.1 - It isn't 100% clear to me where this findings come from - are they from the needs assessment interviews and focus groups? both rounds? I would add a topic sentence to make this clear to the reader.
Table 2. How specificially did the qualitative data inform the design of the initative? Did you decide to focus on space, funding, and leadership training as the three core components of CARES because that is what you heard in the needs assessment or did this qualitative data serve to validate existing plans?
Line 320 - please list number of community and academic partners separately
Figure 3 - it looks like CARE went far beyond the target neighborhoods in the first table. Can you call out how well you've been able to reach community members in those specific neighborhoods. Do you have members that don't have an organizational affiliation? How are they captured in the map? Do you have any demographics information about your members or partners that you could include?
Line 339 - wow! 187 workshops! that's so many! Can you share a bit more somewhere about the staffing of their program to hold that many workshops in such a short period of time? How many staff? Who conducts the workshops?
Line 350 - not sure why you say "only" this seems like a solid number of projects over 2 years' time.
Love the table with the example projects! Do you have any of the evaluation data you described to present?
Reviewer 3 Report
Comments on “Building Capacity for Community-Academia Research Partner- 2 ships by Establishing a Physical Infrastructure for Community 3 Engagement: Morgan CARES”
General Comments:
1. This is a well written and informative article about applying CBPR in practice to low income communities in Baltimore. The processes used to build the positive interactions between the academic program and the local communities are described in detail. They provide a very good description of implementing CBPR in practice.
2. However, the article is weak in the presentation of specific outcomes. A list of programs developed is provided. Other than a noting evaluations are both qualitative and quantitative and a couple of examples such as a Community Rewards Program and a community center which supported people during the COVID-19 pandemic, neither of which is described in detail, it is difficult to identify the outcomes of this program.
3. My suggestion is to rewrite the Discussion. Rather than detailing the justification for success, give examples of the successes and describe how these successes were evaluated. Concrete examples will make the previous descriptions more real and translatable.
4. In addition, it would be useful to highlight the challenges and the resolutions or continuing challenges in implementing this program.
Specific Comments:
1. Key words—Building capacity and Underserved communities are not helpful. Key words enable the reader to focus on specific aspects of the research rather than broad categories.
2. Lines 90-153 would be better placed in a section entitled Background.
3. Please give some details about the “Engaged for Equity CBPR conceptual model”.
Round 2
Reviewer 3 Report
Thank you for responding to my suggestions. I find most responses have covered my comments. I do have one concern however It focuses on the "Engaged for Equity CBPR conceptual model".
I think a bit more information about the model needs to be added. Because this is the framework on which the approach is being assessed ,it is necessary to present the indicators for the assessment. There is no need for details but only to highlight what outcomes are being examined to decide how the program is progressing.
